# “It Happened to Me and It’s Serious”: Conditional Indirect Effects of Infection Severity Narrated in Testimonial Tweets on COVID-19 Prevention

**DOI:** 10.3390/ijerph20136254

**Published:** 2023-06-29

**Authors:** Juan-José Igartua, Laura Rodríguez-Contreras, Íñigo Guerrero-Martín, Andrea Honorato-Vicente

**Affiliations:** Department of Sociology and Communication, Faculty of Social Sciences, Campus Unamuno (Edificio FES), University of Salamanca, 37007 Salamanca, Spain; laurarodriguezcontreras@usal.es (L.R.-C.); i.guerrero@usal.es (Í.G.-M.); andreahonorato@usal.es (A.H.-V.)

**Keywords:** narrative persuasion, COVID-19, testimonial messages, Twitter, health communication, cognitive processes

## Abstract

The health crisis caused by COVID-19 resulted in societal breakdowns around the world. Our research is based on determining which features of testimonial messages are most relevant in increasing persuasive impact. An online experiment with a 2 (severity infection narrative: low vs. high) × 2 (infection target: narrative’s protagonist vs. protagonist’s father) between-subject factorial design was carried out. Young people between 18 and 28 years (N = 278) were randomly assigned to one of the four experimental conditions, where they were asked to read a narrative message in the form of a Twitter thread describing a COVID-19 infection (with mild or severe symptoms) that affected either the protagonist of the message (a 23-year-old young person) or their father. After reading the narrative message, the mediating and dependent variables were evaluated. A message describing a severe COVID-19 infection affecting their protagonist to increase the perception of personal risk increased the persuasive impact through an increase in cognitive elaboration and a reduction in reactance. Our study highlights that creating persuasive messages based on social media targeted at young people that describe a careless behavior resulting in a severe COVID-19 infection can be an appropriate strategy for designing prevention campaigns.

## 1. Introduction

Testimonial messages have become a popular strategy for designing prevention campaigns in the field of health communication [1,2,3,4]. A testimonial message describes a personal story in which the protagonist comments on their situation or experience in a health-related context. For example, the CDC in the United States, in its “Tips From Former Smokers” campaign (https://www.cdc.gov/tobacco/campaign/tips/index.html, accessed on 12 April 2021) developed a series of testimonial messages (written and in video format) where former smokers with different profiles or characteristics described how they started smoking and the consequences this had on their own health, highlighting the problems they personally suffered from (e.g., cancer). In this case, and in other similar cases (as well as in connection with other risky behaviors), the consequences of unhealthy behaviors (such as smoking) on the appearance of health problems (for example, developing cancer) in the person who is the protagonist of the message are described. Additionally, a loss-framing approach is adopted, seeking to elicit negative emotions in the message audience (such as fear or worry) and induce a greater perception of personal risk (“if I smoke, I may get sick”), activate the severity perception (“smoking has serious consequences for health”) and stimulate self-protective behaviors (for example, “I will try to quit smoking in the next three months”).

However, our work focuses on a different context: COVID-19 prevention. The health crisis caused by COVID-19 resulted in societal breakdowns around the world, infecting to date more than 700 million people and causing more than 6 million deaths worldwide [5]. Due to its rapid transmission, infection rate and mortality in those who contracted it, the World Health Organization (WHO) declared a pandemic situation on 11 March 2020 [6]. Without a vaccine available, the virus could only be curbed through changes in daily behaviors (quarantine) and social coordination (social distancing, the use of masks, disinfection of buildings and mobility and transportation restrictions) to reduce the number of COVID-19 infections and avoid overloading healthcare systems [7,8,9].

Some of the earliest research during the pandemic showed that there were large differences in people’s ability to adopt measures to reduce virus transmission [10]. In this sense, it can be stated that COVID-19 has posed a communication challenge on how to approach prevention. It is well known that COVID-19 infection occurs when an infected person comes into contact with an uninfected person [11]. For example, a nurse working in a hospital (lacking appropriate equipment) could become infected simply by being in contact with an infected person whom they have to attend to. It follows that many of the COVID-19 prevention measures were based on social distancing, asking the population to avoid risky situations such as attending celebrations, social gatherings or large events (such as going to the theater or cinema). Moreover, contagion could occur, even if people protected themselves by reducing social contact, as long as there was a person in their social network who did not follow this pattern. Thus, a young person living with their parents could act as a vector of contagion within their family if they did not follow the recommendation to avoid social contact. For example, if this young person went to a party with friends and became infected, they could later infect their parents.

The challenge posed by COVID-19 at the level of medical treatment was, therefore, joined by the challenge faced by health communication professionals in designing prevention campaigns. Specifically, one of the most critical elements was to determine how health campaigns should be designed to raise awareness and encourage people to avoid risky social contacts or to refrain from attending celebrations, parties or similar events in enclosed spaces. The work presented here contributes to health communication research and focuses on analyzing the most effective strategies for preventing COVID-19, a viral infection that is easily transmitted through social contact. Given that COVID-19 spreads when an infected person exhales droplets and very small respiratory particles containing the virus [11], it is vitally important to avoid contact with infected people, knowing that many people may have the virus and not show symptoms. Therefore, it is recommended to avoid crowded places and maintain a social distance.

We present the results of an online experiment in which testimonial messages for the prevention of COVID-19 were created by manipulating two independent variables. The testimonial message, written in the first person and designed as a Twitter “thread”, featured a young man who indicated having celebrated their birthday by organizing a party with their friends without adopting any preventive measures. As a consequence of such a party, a COVID-19 infection affecting the young person themselves or a relative with whom they lived, their father (target of the infection), was mentioned. On the other hand, the narrative described the symptoms experienced by the infected person as mild or severe (severity of infection). In this context, the current study draws on research on narrative persuasion, focusing specifically on the role of identification with the protagonist, narrative transportation, reactance and cognitive elaboration. The aim is to shed light on the psychological mechanisms underlying the impact of message characteristics on preventive measures, including the perceived personal risk of contracting COVID-19, perceived severity of the disease and intentions to adopt preventive behaviors.

## 2. Narrative Persuasion with Testimonial Messages: Advancing the Study of Psychological Mechanisms

Empirical evidence has demonstrated the effectiveness of narrative messages for health communication [1,12,13,14]). Narrative messages represent a communication format consisting of a temporal sequence of causally linked events involving one or more characters [15]. Meta-analysis reviews have found that narrative messages produce significant effects on attitudes, beliefs, behavioral intention and behaviors, although significant variation in these effects has also been observed [16,17]. This means that it is necessary to determine which features of narrative messages are most relevant for eliciting a series of psychological processes that ultimately lead to an impact on measures related to risk perception, severity perception and the adoption of preventive behaviors.

The main theoretical models of narrative persuasion [18,19,20] have established that narrative messages induce persuasive effects on individuals through a set of mechanisms, such as identification with the protagonist, narrative transportation and reactance [18,19,20].

Identification with the protagonist is defined as a multidimensional construct linked to emotional empathy, cognitive empathy, the feeling of merging with the character and adopting their goals [21,22]. It constitutes a psychological phenomenon whereby audience members mentally adopt the perspective of the protagonist of the narrative [23]. Identification allows individuals to overcome the natural tendency to limit their thoughts and feelings to a single perspective [21]. Therefore, research in this field has focused on determining the factors that increase identification with the protagonist of a message, as this can impact the persuasive effectiveness of messages [24]. In this sense, it has been found that characters with positive attributes generate greater identification than characters with negative attributes [25,26].

In the context of our research, the testimonial message manipulated who was infected with COVID-19 (either the young protagonist narrating the story or their father). However, both versions of the story mentioned that the infection had resulted from the protagonist’s imprudent behavior. Engaging in imprudent behavior that could result in a family member’s illness can be regarded as a negative attribute of the story’s protagonist. Therefore, it could be expected that identification with the protagonist of the message would be lower when the person infected by COVID-19 was the protagonist’s father, especially when the symptoms caused by the disease were severe. This led to the first hypothesis of the study:

**H1.** 
*When the narrative describes that the person infected with COVID-19 is the protagonist’s father, there will be lower identification with the protagonist, especially when the symptoms caused by the illness are severe.*


Narrative transportation is a psychological process that involves a state of immersion in the story and integrates three components: the focusing of attention on the story, affective induction and the formation of highly vivid mental images of the characters and situations described in the message. It is also one of the main mechanisms that explain how stories persuade people [27]. Narrative transportation can be influenced by the emotional tone of the message [24]. Indeed, narrative messages are especially powerful when they evoke strong emotions, as narrative transportation is affected by the emotional tone of the message [28]. In the context of the present study, the severity of symptoms associated with COVID-19 described in the testimonial message was manipulated. It could be expected that the version describing severe symptoms (such as high fever and difficulty breathing) compared to the one that described mild symptoms (such as “some cough” or mild headache) would trigger greater emotional engagement that would manifest itself in experiencing greater narrative transportation. This led to the second hypothesis of the study:

**H2.** 
*The narrative message describing a COVID-19 infection with severe symptoms will induce a higher degree of narrative transportation compared to the one describing mild symptoms.*


Reactance is a process associated with resistance to persuasion attempts that is activated when the individual perceives that their freedom of choice is being threatened [29,30]. Moyer-Gusé [19] indicates that narrative messages elicit low levels of reactance because people become immersed in the story. Thus, reducing reactance weakens any critical stance or attitude towards the message, leading to an effective persuasive impact. However, in a recent study on the acquisition of preventive behaviors during the pandemic, it was observed that a greater perception of threat to freedom was linked to higher reactance, which, in turn, was associated with lower levels of adherence to preventive behavior in the face of COVID-19 [31]. In this context, it has been suggested that guilt is a mechanism that holds great persuasive power [32]. Guilt is also linked to transgression and the negative effects that irresponsible or inappropriate behavior can have on others. However, it is important to explore the conditions under which the impact of guilt-based appeals leads to persuasion and their relationship to reactance. In this regard, Bessaravoba et al. [33] conducted a study with high school students (aged 16–18 years) and observed that persuasive messages designed to induce guilt resulted in a decrease in their effectiveness by increasing reactance. In fact, the authors of this study concluded that the use of guilt-based appeals in media campaigns aimed at adolescents could be counterproductive.

Taking as reference the role of guilt and reactance and their effect on the younger population, we proposed that reactance may be an important process in the prevention of COVID-19 through testimonial messages. We assumed that a narrative that attributed to the protagonist the responsibility for the COVID-19 infection of a family member (the protagonist’s father) would induce greater reactance than a narrative that alluded that the target of the contagion was the person who committed the irresponsible behavior, and this effect would be greater when the narrative mentioned that the infection symptoms were severe.

**H3.** 
*A narrative describing that the person infected with COVID-19 is the protagonist’s father will produce higher reactance, especially when the infection is described as severe.*


According to the extended parallel processing model (EPPM), which has previously been applied to explain the effects of fear appeals, the degree to which a person feels threatened by a health problem determines their motivation to act preventively. Thus, the greater the perceived threat, the greater that motivation [34,35,36]. In the context of our research, we assumed that the narrative message describing a severe COVID-19 infection would induce a greater perception of severity than the message describing a mild infection, especially when the contagion target was the protagonist of the story. In a certain sense, a narrative message that alluded to a severe infection and that the person suffering from the disease was the protagonist of the story could be considered a fear appeal and, for this reason, we assumed that this could increase the perceived severity of the infection narrated in the testimonial. In this context, the fourth hypothesis was established:

**H4.** 
*The perceived severity of the infection narrated in the message will be higher when a severe (versus mild) infection is described and the target of the contagion is the protagonist of the story (versus their father).*


A recent study found that the components of the EPPM (perceived efficacy and perceived threat) positively influenced the intention to stay home during the COVID-19 pandemic [37]. Likewise, that study also found that the relationship between the perceived efficacy of the preventive response and the intention to stay home was moderated by perceived threat. Thus, people who assessed the virus as more severe and perceived preventive behaviors as effective in avoiding virus spread were more likely to adopt preventive behaviors against COVID-19. Similar results have been obtained in other studies that highlight the relevance of the perceived severity of the infection in the prevention of COVID-19 [38,39]. These findings suggest that messages that increase the perceived severity of the infection narrated in the message can motivate people to engage in preventive behaviors for themselves, thus, reducing the spread of the virus (see also [40]).

Therefore, we assumed that for the message to exert a significant effect on prevention measures, it had to elicit the perception that the COVID-19 infection narrated in the message was serious and that the person suffering from it was the young protagonist. This process, in turn, would activate different *routes* to explain the persuasive impact: through identification with the protagonist, through narrative transportation and through reactance. However, in this work, we considered a fourth mechanism: cognitive elaboration [41,42,43]. Cognitive elaboration is defined as a process of reflection on the content of the message and constitutes a measure of the intensity of such reflection during the reception process [44]. Cognitive elaboration is the mechanism that has been least explored to date in narrative persuasion research in health communication (see [45]), despite the fact that health messages are designed to stimulate active cognitive processing in individuals. In this sense, we ventured that the experience narrated in the message could serve as an inspiration [46] and stimulate deep cognitive processing in people so that they questioned their previous opinions and adjusted their attitudes towards the pandemic and prevention behaviors.

At this point, we held the assumption that in order for a narrative message to have a persuasive impact, it had to trigger a sequence of psychological processes that acted as mediator mechanisms. In this context, the proposed relevant mediators were the perception of the severity of the infection described in the message (primary mediator), identification with the protagonist, narrative transportation, reactance and cognitive elaboration (secondary mediators). In this given context, it was proposed that when the testimonial narrative described a severe COVID-19 infection (as opposed to a mild one), in which the person telling the story was directly involved (the young protagonist versus their father), the perceived severity of the infection would increase. This heightened perception, in turn, would be associated with a greater identification with the protagonist, increased narrative transportation, lower reactance and greater cognitive elaboration. In turn, these processes would be associated with a greater perceived personal risk of contracting COVID-19, a greater perceived severity of the disease and a greater intention to engage in preventive behavior. Therefore, our fifth hypothesis proposed a moderated serial–parallel mediation model (see Figure 1):

**H5.** 
*It is hypothesized that there will be significant indirect effects on the severity of the infection narrative on several outcomes related to COVID-19 prevention, namely, perceived personal risk of COVID-19 infection (H5a), perceived severity of the disease (H5b) and intention to engage in preventive behavior (H5c). These effects are expected to be mediated by several psychological processes, namely, the perception of severity of the infection narrated in the story, identification with the protagonist, narrative transportation, reactance and cognitive elaboration. However, such indirect effects will only manifest when the infected individual is the protagonist of the story.*


## 3. Method

### 3.1. Participants

A total of 435 individuals participated in the present study. However, the sample was reduced to 278 persons who met the following criteria: aged between 18 and 28, residing in Spain and not having had a past or current COVID-19 infection. In addition, the final selection of the sample also took into account a series of quality controls that were described below (see Section 3.2). Convenience sampling was used by distributing the link to the experimental questionnaire designed with Qualtrics through major media social platforms (Facebook, Twitter, Instagram, WhatsApp) and via email. The link to the experimental questionnaire remained active from 19 May to 1 June 2021. Participants who were included in the final sample had a mean age of 23.14 years (SD = 2.90), with 70.5% being women, 27.7% being men and 1.8% indicating another response (see Table 1).

### 3.2. Design and Procedure

An online experiment with a 2 (severity infection narrative: low vs. high) × 2 (infection target: narrative’s protagonist vs. protagonist’s father) between-subject factorial design was carried out. Participants were randomly assigned to one of the four experimental conditions, where they were asked to read a narrative message in the form of a Twitter thread describing a COVID-19 infection (with mild or severe symptoms) that affected either the protagonist of the message (a 23-year-old young person) or their father.

The questionnaire used was divided into three main blocks: pretest measures, the narrative messages in the form of a Twitter thread, and post-test measures. The first block included an introduction and informed consent. In addition, demographic information was requested (gender, age, country of residence) and included measures on the degree of perceived knowledge about COVID-19 (how they would rate their level of knowledge about coronavirus), the level of personal concern about the COVID-19 pandemic situation (to what extent they felt concerned about the current situation of the COVID-19 pandemic), the level of concern for a possible COVID-19 infection (how often they worried about the possibility of contracting coronavirus) and the question “Do any close relatives have or have had the COVID-19?”.

Immediately after completing the pretest measures, participants were randomly assigned to the experimental conditions, which involved reading a testimonial message. Since two aspects of the message were manipulated (severity infection narrative and infection target), four different versions of the same message were constructed. All versions showed a similar beginning and ending, and only the elements related to the severity of the disease symptoms and the infection target varied. The developed messages (see Independent Variables and Stimulus Materials) were between 213 and 219 words in total, and the estimated reading time was calculated to be 90 s (with a text readability tool, https://legible.es, accessed on 19 April 2021).

After reading the testimonial message, participants completed post-test measures to assess the mediating variables (perceived severity infection, identification with the protagonist, narrative transportation, cognitive elaboration, reactance) and dependent variables (perceived personal risk of COVID-19 infection, perceived severity of COVID-19 and protective behavioral intent against COVID-19). In addition, participants were asked a number of questions to evaluate their retention of key details from the narrative, such as the name and age of the protagonist and who in the story had contracted COVID-19.

Since Qualtrics allowed for a series of quality controls to be implemented, the questionnaire was designed in such a way that it could only be completed in a single session. The average duration to complete the entire process (pretest, message reading and post-test) was approximately 10 min (M = 9.93 min, SD = 4.55, *Mdn* = 8.67), and only participants who had completed the questionnaire between 6 and 45 min were considered. In addition, only the results from participants who took between 30 and 180 s to read the message (M = 53.74 s, SD = 20.61, *Mdn* = 48.85) were counted as valid cases. Moreover, only participants who remembered the central details of the testimonial message were considered valid cases.

All materials related to the online experiment (testimonial messages, measures, datasets and syntax files) are available via the Open Science Framework (OSF): https://osf.io/cz3dt/ (accessed on 27 April 2023).

### 3.3. Independent Variables and Stimulus Materials

The two experimental manipulations were applied to a first-person testimonial narrative message. The message was constructed on the Twitter platform, specifically as a Twitter thread consisting of 7 tweets, each no longer than 280 characters. The credibility and realism of the message were maximized by simulating the interface of the Twitter platform and preserving the structure of the social network (see Appendix A in the OSF).

The testimonial was written in an informal and youthful tone to grant the message greater realism and credibility, with particular attention paid to using gender-neutral language that did not provide clues about the protagonist’s gender. In this sense, prior to the experiment, a pilot study was carried out (N = 28) to select a unisex name for the protagonist of the narrative message. A unisex name is one that is valid for both male and female individuals and does not impose a gender identity. The protagonist’s gender was intended to be neutral in order to avoid any gender-based biases in the reception of the narrative. Participants were asked to select the most suitable unisex name from a list of 15 names, taking into account the Spanish context. Based on the results of the pilot study, Alex (35.7%) was selected as the name of the protagonist in the Twitter testimonial narrative.

The testimonial message elaborated for this study had a clear and causal structure, in such a way that a series of events were presented that were connected in space and time (see Table 2). It was narrated in the first person, since previous research had concluded that the use of the first person was more effective than the second- or third-person narrative voice [42,47]. The message’s protagonist was a 23-year-old individual named Alex Sanchez (@sanchez98_alex). The message described a COVID-19 infection due to risky behavior and the resulting consequences.

The protagonist of the testimonial began by stating that, like most of his friends, they thought that coronavirus was nothing to worry about, that it was like a flu, and that was why their social life had changed little in the last few months. They then recounted that they had celebrated a party with friends to celebrate their birthday and had not paid attention to COVID-19 prevention measures (mask use and social distancing). The use of the “party” factor as a trigger for the infection and transmission of the virus was inspired by the short film “Compromiso en 60 Segundos”, directed by Willy Suárez, winner of the first prize in the Cultura Inquieta microfilm contest and which went viral on social media in Spain at the end of 2020 (https://youtu.be/YolGcDOkL7E, accessed on 8 February 2021). In a third Twitter thread, the protagonist narrated that several days after the party, they (or their father) began to feel unwell, and a PCR test confirmed that it was a COVID-19 infection. They then described the symptoms (mild versus severe) that they (or their father) had four days after the COVID-19 infection was confirmed. The mild symptoms mentioned were having a fever, headache, congestion and “some cough”, and it was emphasized that they had been in bed for four days. In contrast, the version referring to severe symptoms mentioned having a very high fever and difficulty breathing; additionally, it was stated that they had been in the hospital for four days and that medical treatment with oxygen had been necessary. The message also indicated that the doctor had said that the infection could vary between the fifth and eighth day, so anything could be expected to happen, thus, creating a scenario of uncertainty. Faced with this situation, the protagonist expressed feeling fear and concern, stating that they thought that all of this could have been avoided, now valuing that COVID-19 was more serious than they had thought, and ending with a prevention message directed at young people: “Take care, protect yourselves, we are not immortal”.

### 3.4. Measures

*Perceived infection target (manipulation check)*. To contrast the effectiveness of the experimental manipulation related to the infection target (who was the person suffering from the COVID-19 infection), two items were included: “the person affected by COVID-19 who narrated the message was a young person” and “the person affected by COVID-19 who narrated the message had become infected due to imprudence” (from 1 = strongly disagree to 7 = strongly agree).

*Perceived severity infection*. An ad hoc scale was created consisting of two items: “the person infected with COVID-19 experienced severe symptoms” and “it is very unlikely that the disease described in the message will eventually endanger the life of the person infected with COVID-19” (from 1 = strongly disagree to 7 = strongly agree). Both items showed a negative correlation (*r*(276) = −0.18, *p* <.001) and were combined (after recoding the second item) to form an index of perceived severity infection (M = 4.90, SD = 1.28). This index was used as an outcome variable to test H4 and as a mediating variable to test H5 (see [48]).

*Identification with the protagonist*. Identification was assessed using an 11-item scale [49] that measured the degree of identification with a specific character (e.g., “I felt as if I were Alex”; from 1 = not at all to 5 = very much). The 11 items were averaged into a reliable scale (α = 0.88, M = 2.87, SD = 0.75).

*Narrative transportation*. It was assessed by means of the transportation scale—short form [50]—which consisted of 5 items (e.g., “I was mentally involved in the narrative while reading it”; from 1 = strongly disagree to 7 = strongly agree). The 5 items were averaged into a reliable scale (α = 0.74, M = 4.18, SD = 1.17).

*Cognitive elaboration*. An adapted version of the cognitive elaboration scale developed by Igartua and Rodríguez-Contreras [42] was used, consisting of four items (e.g., “while reading the narrative, I intensely reflected on the topic of the coronavirus”; from 1 = strongly disagree to 7 = strongly agree). The 4 items were averaged into a reliable scale (α = 0.84, M = 4.66, SD = 1.28).

*Reactance*. This was assessed with the perceived threat to freedom scale created by Shen (2015) [51], comprising 4 items (e.g., “the message was trying to pressure me”; from 1 = strongly disagree to 7 = strongly agree). A reactance index was constructed from calculating the average across the four items (α = 0.84, M = 2.06, SD = 1.29).

*Perceived personal risk of* COVID-19 *infection*. Taking as a reference the study conducted by Jahangiry et al. [52], participants were asked to indicate their degree of agreement or disagreement with the statement “my risk of contracting coronavirus is very high” (from 1 = strongly disagree to 7 = strongly agree; M = 3.34, SD = 1.53).

*Perceived severity of* COVID-19. A scale composed of 5 items (e.g., “I think coronavirus is more severe than influenza”, “COVID-19 can cause serious health problems”; from 1 = strongly disagree to 7 = strongly agree) was developed based on two previous studies [52,53]. A perceived severity of COVID-19 index was constructed from calculating the average across the five items (α = 0.71, M = 6.04, SD = 0.82).

*Protective behavioral intent against* COVID-19. To assess the participants’ protective behaviors, the following question was included (based on the study by Rosero-Bolaños et al. [54]): “on a scale of 0 (not at all likely) to 10 (very likely), what is the probability that you will not attend parties and/or meetings in the next 4 weeks, for fear of infecting others with coronavirus (COVID-19)” (M = 6.46, SD = 2.99).

*Recall of the details of the narrative*. To assess the retention of essential information from the testimonial message, participants were asked the following questions at the end of the questionnaire: a) “What was the name of the person who was the protagonist of the story you read?” (1 = I do not remember; 2 = Dani; 3 = Alex (correct); 4 = Rosa); b) “How old was the protagonist of the message?” (1 = I do not remember; 2 = 18 years old; 3 = 23 years old (correct): 4 = 28 years old); c) “who was the person who, according to the message, had health problems after being infected with coronavirus?” (1 = I do not remember; 2 = the person who narrated the message; 3 = a friend of the person who narrated the message; 4 = the father of the person who narrated the message; 5 = the mother of the person who narrated the message).

### 3.5. Data Analysis

Data analyses were conducted using IBM SPSS 28 statistical software. Descriptive analyses were calculated to examine the sample demographics and basic statistical information of the measures (means and standard deviations). The internal consistency reliability was calculated for all self-report scales composed of 3 or more items using Cronbach’s alpha coefficient. A one-way analysis of variance (ANOVA), Student’s t-test and chi-squared test were used to check for successful randomization and to test the efficacy of experimental manipulations. Bivariate correlations between the mediating and dependent variables were analyzed with Pearson’s product moment coefficient of correlation. A factorial ANOVA was performed to determine the impact of the severity of the infection on narrative transportation (H2), including the infection target as a second independent variable. To test hypotheses 1, 3, 4 and 5, the PROCESS macro (version 4.3) for SPSS was used. This macro made it possible to test different moderation, mediational and conditional process models [55]. To carry out the analyses with PROCESS, the independent variables (severity infection and infection target) were coded −0.5 (for “low-severity infection narrative” and for “infection target: protagonist’s father”, respectively) and 0.5 (for “high-severity infection narrative” and for “infection target: narrative’s protagonist”, respectively). This coding approach is called the main effect parameterization because of the use of −0.5 and 0.5 for coding the values of the focal predictor and the values of the moderating variable that produces regression coefficient estimates that correspond to the main effects of each independent variable from a 2 × 2 ANOVA [56]. To test H1, H3 and H4, the PROCESS macro was used, applying model 1 (simple moderation). To test H5 (a moderated serial–parallel mediation model), the PROCESS macro was used, applying a customized conditional process model (10,000 bootstrapping samples to generate 95% confidence intervals using the percentile method). According to the bootstrapping method, an indirect effect is statistically significant if the confidence interval established (CI at 95%) does not include the value 0. If the value 0 is included in the CI, the indirect effect is equal to 0, that is, there is no association between the variables considered.

## 4. Results

### 4.1. Preliminary Analysis

The randomization was successful: the conditions did not differ significantly on gender (χ^2^(6, N = 278) = 3.37, *p* = 0.761), age (F(3, 274) = 1.65, *p* = 0.287), duration (in minutes) of participation in the experiment (F(3, 274) = 0.29, *p* = 0.839), reading time of the testimonial message (F(3, 274) = 0.38, *p* = 0.764), perceived knowledge about COVID-19 (F(3, 274) = 1.65, *p* = 0.178), level of personal concern about the COVID-19 pandemic situation (F(3, 274) = 0.09, *p* = 0.961) and level of concern for a possible COVID-19 infection (F(3, 274) = 0.27, *p* = 0.844). There were also no statistically significant differences between experimental conditions in the percentage of participants with close family members who currently had, or have had in the past, COVID-19 infection. (χ^2^(3, N = 278) = 1.13, *p* = 0.769).

The experimental manipulation of the severity of the infection was effective. Participants exposed to the high-severity message (M = 5.44, SD = 1.36) showed a higher degree of agreement with the statement “the person infected with COVID-19 experienced severe symptoms” than those exposed to the low-severity message (M = 4.34, SD = 1.63; t(255.89) = −6.03, *p* < 0.001; Levene’s test F = 11.96, *p* < 0.001). In addition, persons exposed to the low-severity narrative showed a higher degree of agreement (M = 3.37, SD = 1.78) with the statement “it is very unlikely that the disease will end up endangering the life of the COVID-19 infected person referred to in the message” compared to persons exposed to the high-severity narrative (M = 2.86, SD = 1.65; t(276) = 2.49, *p* = 0.007).

The experimental manipulation of the infection target was also effective. Participants exposed to the message referring to the protagonist’s infection showed a higher degree of agreement with the statement “the person affected by COVID-19 who narrates the message was a young person” (M = 6.77, SD = 0.68) than those exposed to the message referring to the protagonist’s father’s infection (M = 5.03, SD = 2.17; t(176.07) = −9.20, *p* < 0.001; Levene’s test F = 220.18, *p* < 0.001). Moreover, those exposed to the message alluding to the protagonist’s infection showed a higher degree of agreement with the statement “the person affected by COVID-19 who narrates the message had been infected by an imprudent action” (M = 6.36, SD = 1.10) than those exposed to the message alluding to the protagonist’s father’s infection (M = 5.71, SD = 1.69; t(251.64) = −3.81, *p* < 0.001; Levene’s test F = 17.77, *p* < 0.001).

Correlations between the mediating variables and the dependent variables were also analyzed (see Table 3). This analysis confirmed that the mediating processes showed convergent correlations with each other. For example, the perceived severity of the infection was associated with greater cognitive processing. In addition, the mediating processes were also significantly associated with the dependent variables (e.g., a positive and significant correlation was observed between cognitive elaboration and perceived personal risk).

### 4.2. H1: The Impact of the Infection Target on Identification with the Protagonist Is Moderated by the Severity of the COVID-19 Infection

H1 posited that the narrative alluding to a COVID-19 infection of the protagonist’s father would induce less identification with the protagonist than the narrative alluding to an infection on the protagonist himself, specifically when the infection was described as severe. The PROCESS macro (Model 1) was used to test this hypothesis of moderation, which yielded a non-statistically significant interaction effect (B = 0.11, SE = 0.18, *p* = 0.533). In addition, neither statistically significant main effects of the infection target (B = −0.11, SE = 0.09, *p* = 0.216) nor of the severity infection described in the message (B = 0.13, SE = 0.18, *p* = 0.620) were observed on identification with the protagonist. Therefore, H1 did not receive empirical support.

### 4.3. H2: Narrative Describing an Infection with Severe Symptoms Induces Greater Narrative Transportation Than the One That Describe Mild Symptoms

H2 proposed that a message mentioning a severe COVID-19 infection would result in greater narrative transportation compared to a message referring to a mild infection. Although no main effect was predicted for the infection target (nor an interaction effect between the two independent variables), an ANOVA was conducted regardless. The results showed that the severity of the COVID-19 infection did not significantly influence the degree of narrative transportation (F(1, 274) = 0.63, *p* = 0.427). The message describing severe symptoms (M = 4.23, SE = 0.09) induced a similar level of narrative transportation as the message describing mild symptoms (M = 4.12, SE = 0.10). Thus, H2 did not receive empirical support. There was no statistically significant main effect observed for the infection target (F(1, 274) = 3.34, *p* = 0.068) nor an interaction effect between the two independent variables (F(1, 274) = 1.11, *p* = 0.291).

### 4.4. H3: The Impact of the Infection Target on Reactance Is Moderated by the Severity of the COVID-19 Infection

According to H3, the narrative mentioning that the person infected with COVID-19 was the protagonist’s father would induce greater reactance compared to the narrative mentioning that the infected person was the young person, especially if the infection was described as severe. The PROCESS macro (model one) was used to test this hypothesis of moderation, which yielded a non-statistically significant interaction effect (B = −0.32, SE = 0.31, *p* = 0.292). In addition, neither statistically significant main effects of the infection target (B = −0.00, SE = 0.15, *p* = 0.994) nor of the severity of the infection described in the message (B = 0.16, SE = 0.15, *p* = 0.292) were observed on reactance. Therefore, H3 was not empirically supported.

### 4.5. H4: The Effect of the Severity of the COVID-19 Infection on Perceived Severity Is Moderated by the Infection Target

H4 predicted that the narrative message describing a severe infection would lead to a greater perceived severity, especially when the infection target was the protagonist of the story. Therefore, we expected to find an interaction effect between the severity of the infection described in the message and the infection target on the perception of the severity of the infection narrated in the story. The PROCESS macro (Model 1) was used to test this hypothesis of moderation, which yielded a statistically significant interaction effect (B = 0.90, SE = 0.28, *p* = 0.001). The conditional effect analysis (see Figure 2) revealed that the severity of the infection symptoms described in the message significantly increased the perceived severity only when the infection target was the narrative’s protagonist (θ _X_ → _Y | (infection target = “narrative’s protagonist)_ = 1.25, SE = 0.20, *p* < 0.001), but not when it was the protagonist’s father (θ _X_ → _Y | (infection target = protagonist’s father)_ = 0.34, SE = 0.19, *p* = 0.073). In conclusion, the severity of the infection increased the perceived severity of the infection narrated in the message only when the infection target was the protagonist of the testimonial. These results provided empirical support for H4.

### 4.6. H5: Testing a Moderated Serial–Parallel Mediation Model

H5 proposed a moderated serial–parallel mediation model. This model predicted indirect (specific and conditional) effects of the severity of the infection described in the message on the perceived personal risk of COVID-19 infection (H5a), on the perceived severity of COVID-19 (H5b) and on the intention of preventive behavior (H5c) mediated by the perception of the severity of the infection narrated in the story (primary mediator), identification with the protagonist, narrative transportation, reactance and cognitive elaboration (secondary mediators). However, it was predicted that these indirect effects would only manifest themselves when the infected person was the protagonist of the message. To test this model (which included five mediating variables, one independent variable and one moderating variable), the PROCESS macro (customized model) was used.

The results of the analyses provided partial support for H5, as cognitive elaboration and reactance constituted relevant mediating mechanisms for explaining the impact of the severity of the COVID-19 infection narrated in the message on the outcome variables. However, neither identification with the protagonist nor narrative transportation constitute significant mediating mechanisms. The results were presented individually for each dependent variable in the form of tables (where the specific conditional indirect effects were reported) and figures (where the unstandardized regression coefficients quantifying the relationship between the different variables were reported).

Regarding the first dependent variable considered (perceived personal risk of COVID-19 infection), it was observed that when the testimonial narrative referred to a severe infection in which the person narrating the story was involved (the young protagonist who had engaged in risky behavior), the perceived severity infection increased (B = 0.90, SE = 0.28, *p* = 0.001). In turn, the perceived severity infection narrated in the message was associated with greater narrative transportation (B = 0.16, SE = 0.05, *p* = 0.003), greater cognitive elaboration (B = 0.23, SE = 0.05, *p* < 0.001) and lower reactance (B = −0.15, SE = 0.05, *p* = 0.010). However, only cognitive elaboration showed a significant effect on perceived personal risk of contracting COVID-19 (B = 0.19, SE = 0.08, *p* = 0.014). The indirect effect of the severity of the infection narrated in the message on perceived personal risk, through the perceived severity of the infection and cognitive elaboration (serial mediation), was statistically significant only when the infection target was the young protagonist (effect = 0.0575, SE = 0.0298, 95% CI: 0.0071, 0.1234) (see Table 4 and Figure 3a).

In relation to the second dependent variable, it was observed that both cognitive elaboration (B = 0.13, SE = 0.04, *p* = 0.001) and reactance (B = −0.15, SE = 0.03, *p* < 0.001) showed significant effects on the perceived severity of COVID-19. Additionally, two significant conditional specific indirect effects were obtained through the mediation of cognitive elaboration (effect = 0.0398, SE = 0.0186, 95% CI: 0.0102, 0.0820) and reactance (effect = 0.0296, SE = 0.0156, 95% CI: 0.0054, 0.0652), in both cases when the infection target was the young protagonist (see Table 5 and Figure 3b).

Finally, concerning the third dependent variable, it was also observed that both cognitive elaboration (B = 0.32, SE = 0.15, *p* = 0.041) and reactance (B = −0.35, SE = 0.13, *p* = 0.009) showed a significant effect on protective behavioral intent against COVID-19. In addition, two specific conditional indirect effects were observed through the mediation of cognitive elaboration (effect = 0.0946, SE = 0.0608, 95% CI: 0.0006, 0.2365) and reactance (effect = 0.0690, SE = 0.0384, 95% CI: 0.0079, 0.1557), in both cases when the infection target was the young protagonist (see Table 6 and Figure 3c).

## 5. Discussion

The pandemic caused by COVID-19 has posed a global challenge that has required the coordinated action of local, state and international institutions. The main challenge during the pandemic was to reduce the number of infections and cases of COVID-19 to avoid overwhelming healthcare systems [57]. The second major challenge was biomedical in nature: it was necessary to develop an effective vaccine to curb the pandemic and protect the entire population [58]. Finally, the third challenge encompassed all individuals and institutions that aimed to prevent COVID-19 by increasing the awareness of modes of transmission and preventive strategies [59]. This last challenge is closely linked to communication activities, the dissemination of scientifically validated information and the design of persuasive strategies to convince the population to adopt infection prevention practices (such as social distancing, mask wearing, building disinfection and mobility and transportation restrictions) [7,8,9].

This paper focused precisely on the communicative challenge posed by the pandemic to stimulate prevention behaviors. Building on research on narrative persuasion and, partially, on the impact of fear appeals (see [60]), an experiment was designed using testimonial messages in the form of a Twitter “thread” as a stimulus. Social media played a decisive role during the pandemic. Social networks such as Twitter constituted a primary platform for disseminating health information from institutions [61]. However, they were also used to spread fake news, conspiracy theories denying the existence of the virus and information opposing the use of masks or social distancing measures [62,63,64]. Nonetheless, we acknowledged that Twitter could also serve as an optimal platform for sharing testimonial messages for raising awareness around prevention measures during the pandemic. We consider that testimonial messages disseminated through social networks such as Twitter could constitute narrative vaccines [49,65] or narrative pills [66] that spread personal stories that facilitated a persuasive impact, especially among the younger population. In our case, the testimonial message was led by a young person who narrated a risky behavior (holding a party without taking COVID-19 prevention measures) that resulted in a COVID-19 infection that affected either the young person or a family member with whom they lived (contagion target). Furthermore, the narration described the symptoms experienced by the infected person as either mild or severe (infection severity). We opted for this narrative persuasion approach because awareness messages directed at young people led by authority figures (such as healthcare professionals or institutional actors) can provoke rejection from this group [67].

The experiment, carried out with a convenience sample (N = 278) of young people aged 18 to 28 years, allowed us to advance our knowledge on the psychological mechanisms that explained the impact of two message features (the severity of the infection described in the testimonial and the infection target) on preventive measures, such as the perceived personal risk of contracting COVID-19, the perceived severity of COVID-19 and the intention to engage in preventive behavior. Five hypotheses were established, although only two of them received empirical support.

The first hypothesis predicted that when the narrative described the person infected with COVID-19 as the protagonist’s father, there would be less identification with the protagonist of the message, especially when the symptoms caused by the disease were severe. This hypothesis was based on research on the (positive or negative) characteristics of narrative message protagonists and their effects on identification [24]. In this context, engaging in an imprudent behavior (organizing a party and not wearing a mask) that could potentially result in a family member becoming ill was considered a negative trait of the protagonist of the story. However, our hypothesis did not receive empirical support, since no statistically significant interaction effect was observed between the two manipulated independent variables on identification. Moreover, identification was not affected by who was the target of the contagion or by the severity of the symptoms. However, it should be noted that the level of identification with the protagonist of the message was low (M = 2.87, SD = 0.75), placing it below the theoretical midpoint of the scale (value of three; t(277) = −2.68, *p* = 0.004). This may mean that, overall, participants felt that a young person who had acted irresponsibly during the pandemic was not a positive role model to identify with.

The second hypothesis predicted an effect of the severity of the symptoms narrated in the message on narrative transportation. However, this hypothesis did not receive empirical support either. Nonetheless, in this case, it was observed that, overall, the testimonial message induced a level of narrative transportation (M = 4.18, SD = 1.17) above the theoretical midpoint of the scale (value of four; t(277) = 2.58, *p* = 0.005). The third hypothesis also did not receive empirical support, as the narrative mentioning that the person infected with COVID-19 was the father and that the infection was severe did not induce the highest level of reactance. Again, it was observed that, overall, the level of reactance experienced by the participants was low (M = 2.06, SD = 1.29), below the theoretical midpoint of the scale (value of four; t(277) = −24.98, *p* < 0.001). This result was convergent with Moyer-Gusé’s [19] entertainment overcoming resistance model (EORM), which proposed that narrative messages have the capacity to reduce reactance, given that they are not perceived as aiming to persuade.

The fourth hypothesis did receive empirical support. It was hypothesized that a narrative message describing a serious infection affecting the protagonist of the story could be considered a fear appeal. Therefore, it was considered that such a combination of elements in the message would elicit a heightened perception of the severity of the infection described in the testimonial. The results were consistent with this prediction, which was in line with the postulates of the EPPM [36] on the effects of fear on the activation of protective motivation against a particular disease.

Finally, a moderated serial–parallel mediation model was proposed. It was considered that for a narrative message to have a persuasive impact on measures related to COVID-19 prevention, it had to stimulate a series of psychological processes acting as mediating mechanisms. The analysis of the specific conditional indirect effects showed that both cognitive elaboration and reactance were relevant mechanisms to explain the impact of the characteristics of the message on the dependent variables considered (perceived personal risk of contracting COVID-19, perceived severity of the disease, and intention to engage in preventive behavior). Therefore, it was noted that for a message describing a severe COVID-19 infection affecting their protagonist to increase the perception of personal risk, the perception that COVID-19 is a serious disease and that preventive action is necessary (not attending parties or gatherings for fear of contracting coronavirus), a dual process had to be activated. Thus, two routes were established that would explain the persuasive impact, first through activating the perception that the infection described in the message was severe and the subsequent increase in cognitive processing (e.g., “reading the message has made me think deeply about measures to prevent the transmission of the coronavirus”) and second through the activation of the perception that the infection narrated in the message was severe and the subsequent reduction in reactance (e.g., “the message was trying to manipulate me”). More importantly, the results of this study showed that identification and narrative transportation did not constitute relevant mediating mechanisms.

The present study had two major limitations. Firstly, it did not include any measure of the emotional impact provoked by the reading of the testimonial messages (fear and guilt, in particular). It would have been interesting to contrast the effect of the infection target on the experience of guilt and, subsequently, to analyze the relationship between guilt and reactance. The second limitation of this work was that the proposed mediators were measured rather than experimentally manipulated, which prevented drawing conclusions with complete certainty regarding the proposed causal sequence between the different psychological mechanisms. However, although temporal precedence is an important element for establishing a causal inference, it is also necessary to propose a theoretical argument about the relationship between the mediating mechanisms, a condition that our work fulfilled by relying on theoretical models of narrative persuasion (EORM) [19] and on the EPPM model [36]. Nevertheless, future research should use other methodological approaches to address such causal inference problems [68].

## 6. Conclusions

As a general conclusion, our study highlighted that creating persuasive messages based on social media (Twitter) targeted at young people that describe a careless behavior resulting in a severe COVID-19 infection could be an appropriate strategy for designing prevention campaigns (see also [69,70] on the role of Twitter in the dissemination of medical information and misinformation during the COVID-19 pandemic). Such a message could inspire the audience (by stimulating deep reflection and reducing reactance), leading to the adoption of self-protective and socially responsible prevention measures [46]. The dual mediation model outlined highlighted the need to stimulate the perception of severity, which was congruent with the EPPM model [36,60]. Such a process would act as a catalyst for activating cognitive elaboration and reducing reactance. This model could serve as a blueprint for developing social media campaigns aimed at addressing other health or social issues that young people face today, such as HIV–AIDS prevention, sexually transmitted diseases, alcohol and tobacco consumption and obesity, among others.

## Figures and Tables

**Figure 1 ijerph-20-06254-f001:**
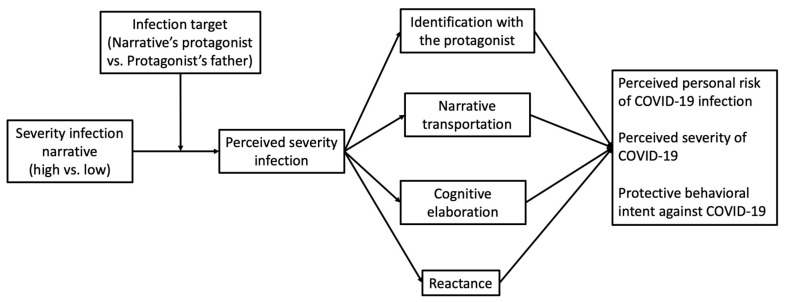
Hypothesized moderated serial–parallel mediation model.

**Figure 2 ijerph-20-06254-f002:**
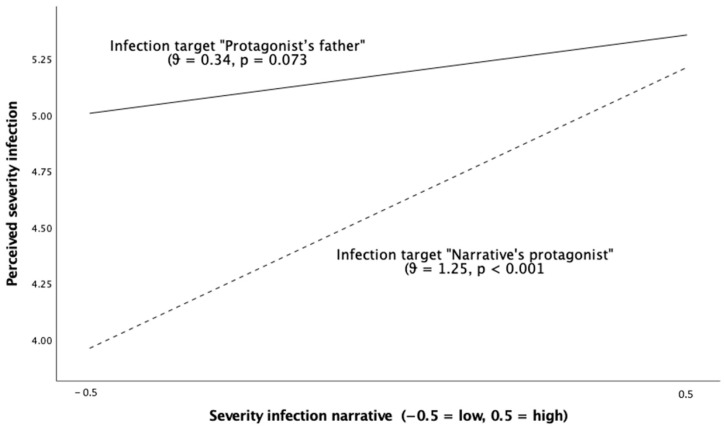
Moderating effect of infection target on the relationship between the severity of the symptoms described in the Twitter narrative and the perceived severity of the infection.

**Figure 3 ijerph-20-06254-f003:**
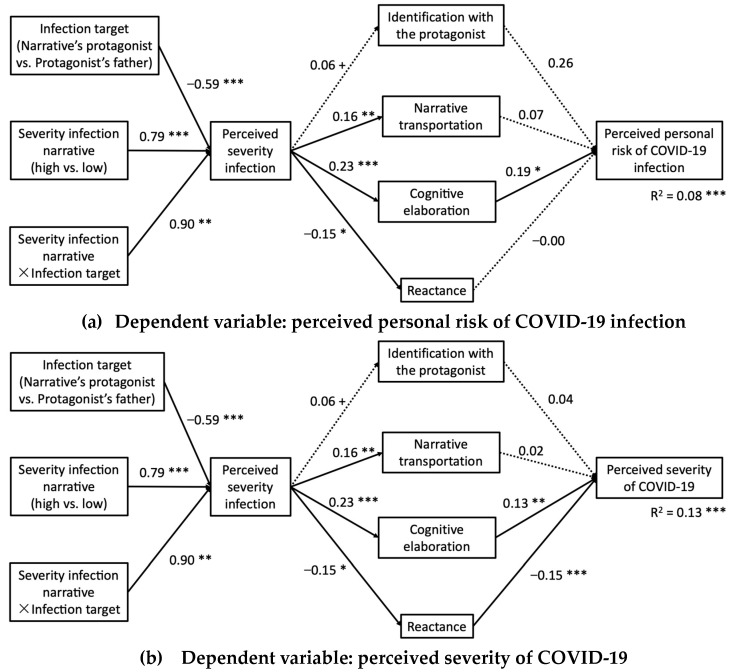
Results of the moderated serial–parallel mediation model (H5). The figures show the non-standardized regression coefficients, *B.* The dashed line represents non-significant coefficients. + *p* a 0.10, * *p* < 0.05, ** *p* < 0.01, *** *p* < 0.001. (**a**) Dependent variable: perceived personal risk of COVID-19 infection. (**b**) Dependent variable: perceived severity of COVID-19. (**c**) Dependent variable: protective behavioral intent against COVID-19.

**Table 1 ijerph-20-06254-t001:** Characteristics of the study participants (N = 278).

Variables	Mean (SD) or Percentage	Range
Age	M = 23.14 SD = 2.90	18–28
Sex	Male: 77 (27.7%) Female: 196 (70.5%) Other response: 5 (1.8%)	
Perceived knowledge about COVID-19	M = 3.25 SD = 0.67	1 (low)–5 (high)
Level of personal concern about the COVID-19 pandemic situation	M = 6.85 SD = 1.75	0 (low)–10 (high)
Level of concern for a possible COVID-19 infection	M = 3.53 SD = 0.94	1 (low)–5 (high)
Do any close relatives have or have had the COVID-19?	No: 147 (52.9%) Yes: 131 (47.1%)	

**Table 2 ijerph-20-06254-t002:** Testimonial messages used as experimental stimuli.

Severity Infection Narrative: Low -Infection Target: Narrative’s Protagonist (215 Words)	Severity Infection Narrative: High -Infection Target: Narrative’s Protagonist (224 Words)
Today I wanted to talk to you about the situation I’m going through at the moment. A thread. Like most of my friends, I had always thought that coronavirus was not something I had to worry about. I thought it was like a flu, nothing more. That’s why my social life had changed little in the last few months. I turned 23 last week and of course, I threw a party to celebrate. We weren’t too many, but we didn’t pay much attention to safety. Being with your friends makes you forget about COVID, masks, distance… Two days after the party I started to feel sick. I had the test, a PCR, and they told me I had COVID. The news broke me. I’ve been in bed for four days, with fever, headache, congestion, and some coughing. Let’s see how the disease evolves in the next few days. The doctor has said that between the fifth and eighth day of the infection is when everything can change and determine if the evolution is going to get worse, so I don’t know what will happen and it scares me. Now I think this could have been avoided. I see this COVID thing is more serious than I thought. Take care, protect yourselves, we are not immortal.	Today I wanted to talk to you about the situation I’m going through at the moment. A thread. Like most of my friends, I had always thought that coronavirus was not something I had to worry about. I thought it was like a flu, nothing more. That’s why my social life had changed little in the last few months. I turned 23 last week and of course, I threw a party to celebrate. We weren’t too many, but we didn’t pay much attention to safety. Being with your friends makes you forget about COVID, masks, distance… Two days after the party I started to feel sick. I had the test, a PCR, and they told me I had COVID. The news broke me. I have been in the hospital for four days, with a very high fever and difficulty breathing, they even put me on oxygen. Let’s see how the disease evolves in the next few days. The doctor has said that between the fifth and eighth day of the infection is when everything can change and determine if the evolution is going to get worse, so I don’t know what will happen and it scares me. Now I think this could have been avoided. I see this COVID thing is more serious than I thought. Take care, protect yourselves, we are not immortal.
**Severity Infection Narrative: Low** **-Infection Target: Protagonist’s Father (218 Words)**	**Severity Infection Narrative: High** **-Infection Target: Protagonist’s Father (226 Words)**
Today I wanted to talk to you about the situation I’m going through at the moment. A thread. Like most of my friends, I had always thought that coronavirus was not something I had to worry about. I thought it was like a flu, nothing more. That’s why my social life had changed little in the last few months. I turned 23 last week and of course, I threw a party to celebrate. We weren’t too many, but we didn’t pay much attention to safety. Being with your friends makes you forget about COVID, masks, distance… Two days after the party my father started to feel sick. He had the test, a PCR, and they told him he had COVID. The news broke me. My father has been in bed for four days, with fever, headache, congestion, and some coughing. Let’s see how the disease evolves in the next few days. The doctor has said that between the fifth and eighth day of the infection is when everything can change and determine if the evolution is going to get worse, so I don’t know what will happen and it scares me. Now I think this could have been avoided. I see this COVID thing is more serious than I thought. Take care, protect yourselves, we are not immortal.	Today I wanted to talk to you about the situation I’m going through at the moment. A thread. Like most of my friends, I had always thought that coronavirus was not something I had to worry about. I thought it was like a flu, nothing more. That’s why my social life had changed little in the last few months. I turned 23 last week and of course, I threw a party to celebrate. We weren’t too many, but we didn’t pay much attention to safety. Being with your friends makes you forget about COVID, masks, distance… Two days after the party my father started to feel sick. He had the test, a PCR, and they told him he had COVID. The news broke me. My father has been in the hospital for four days, with a very high fever and difficulty breathing, they even put him on oxygen. Let’s see how the disease evolves in the next few days. The doctor has said that between the fifth and eighth day of the infection is when everything can change and determine if the evolution is going to get worse, so I don’t know what will happen and it scares me. Now I think this could have been avoided. I see this COVID thing is more serious than I thought. Take care, protect yourselves, we are not immortal.

**Table 3 ijerph-20-06254-t003:** Descriptive analysis and correlations between mediating and dependent variables.

	1	2	3	4	5	6	7	8
1 Perceived severity infection	-	-	-	-	-	-	-	-
2 Identification	0.10 *	-	-	-	-	-	-	-
3 Narrative transportation	0.17 **	0.73 ***	-	-	-	-	-	-
4 Cognitive elaboration	0.23 ***	0.46 ***	0.45 ***	-	-	-	-	-
5 Reactance	−0.15 **	0.02	0.04	−0.10 *	-	-	-	-
6 Perceived personal risk of COVID-19 infection	0.04	0.24 ***	0.22 ***	0.25 ***	−0.01	-	-	-
7 Perceived severity of COVID-19	0.17 ***	0.16 **	0.15 **	0.27 ***	−0.26 ***	0.16 **	-	-
8 Protective behavioral intent against COVID-19	0.02	−0.09 +	−0.13 *	0.06	−0.17 ***	−0.02	0.14 **	-
Mean	4.90	2.87	4.18	4.66	2.06	3.34	6.04	6.45
Standard deviation	1.28	0.75	1.17	1.28	1.19	1.53	0.82	2.99

Note. N = 278. For all the variables, a higher score indicates a greater intensity of the considered process, from 1 = low to 7 = high (except for the identification scale, which has a theoretical range from 1 = low to 5 = high). + *p* < 0.10; * *p* < 0.05; ** *p* < 0.01; *** *p* < 0.001.

**Table 4 ijerph-20-06254-t004:** Results of the conditional specific indirect effects of the severity of the symptoms described in the Twitter narrative on perceived personal risk of COVID-19 infection (H5).

Conditional Specific Indirect Effects	Effect	Boot SE	Boot 95% CI
Severity infection narrative → Perceived severity infection → Identification → Perceived personal risk			
- Infection target: Protagonist’s father	0.0056	0.0065	[−0.0039, 0.0218]
- Infection target: Narrative’s protagonist	0.0201	0.0204	[−0.0098, 0.0701]
IMM = 0.0145 (95% CI: −0.0065, 0578)			
Severity infection narrative → Perceived severity infection → Narrative transportation → Perceived personal risk			
- Infection target: Protagonist’s father	0.0042	0.0090	[−0.0088, 0.0281]
- Infection target: Narrative’s protagonist	0.0152	0.0269	[−0.0320, 0.0782]
IMM = 0.0109 (95% CI: −0.0246, 0.0581)			
Severity infection narrative → Perceived severity infection → Cognitive elaboration → Perceived personal risk			
- Infection target: Protagonist’s father	0.0160	0.0125	[−0.0026, 0.0466]
- **Infection target: Narrative’s protagonist**	**0.0575**	**0.0298**	**[0.0071, 0.1234]**
IMM = 0.0415 (95% CI: 0.0041, 0.0993)			
Severity infection narrative → Perceived severity infection → Reactance → Perceived personal risk			
- Infection target: Protagonist’s father	0.0004	0.0045	[−0.0084, 0.0106]
- Infection target: Narrative’s protagonist	0.0014	0.0134	[−0.0268, 0.0284]
IMM = 0.0010 (95% CI: −0.0200, 0.0203)			

Note. Significant specific conditional indirect effects in bold. IMM = index of moderated mediation (difference between conditional indirect effects).

**Table 5 ijerph-20-06254-t005:** Results of the conditional specific indirect effects of the severity of the symptoms described in the Twitter narrative on perceived severity of COVID-19 (H5).

Conditional Specific Indirect Effects	Effect	Boot SE	Boot 95% CI
Severity infection narrative → Perceived severity infection → Identification → Perceived severity of COVID-19			
- Infection target: Protagonist’s father	0.0010	0.0029	[−0.0030, 0.0088]
- Infection target: Narrative’s protagonist	0.0037	0.0089	[−0.0094, 0.0271]
IMM = 0.0027 (95% CI: −0.0069, 0.0208)			
Severity infection narrative → Perceived severity infection → Narrative transportation → Perceived severity of COVID-19			
- Infection target: Protagonist’s father	0.0015	0.0042	[−0.0059, 0.0116]
- Infection target: Narrative’s protagonist	0.0054	0.0129	[−0.0199, 0.0330]
IMM = 0.0039 (95% CI: −0.0154, 0.0248)			
Severity infection narrative → Perceived severity infection → Cognitive elaboration → Perceived severity of COVID-19			
- Infection target: Protagonist’s father	0.0111	0.0086	[−0.0013, 0.0320]
- **Infection target: Narrative’s protagonist**	**0.0398**	**0.0186**	**[0.0102, 0.0820]**
IMM = 0.0287 (95% CI: 0.0057, 0.0662)			
Severity infection narrative → Perceived severity infection → Reactance → Perceived severity of COVID-19			
- Infection target: Protagonist’s father	0.0083	0.0074	[−0.0010, 0.0273]
- **Infection target: Narrative’s protagonist**	**0.0296**	**0.0156**	**[0.0054, 0.0652]**
IMM = 0.0213 (95% CI: 0.0032, 0.0495)			

Note. Significant specific conditional indirect effects in bold. IMM = index of moderated mediation (difference between conditional indirect effects).

**Table 6 ijerph-20-06254-t006:** Results of the conditional specific indirect effects of the severity of the symptoms described in the Twitter narrative on protective behavioral intent against COVID-19 (H5).

Conditional Specific Indirect Effects	Effect	Boot SE	Boot 95% CI
Severity infection narrative → Perceived severity infection → Identification → Protective behavioral intent against COVID-19			
- Infection target: Protagonist’s father	−0.0026	0.0106	[−0.0294, 0.0171]
- Infection target: Narrative’s protagonist	−0.0094	0.0328	[−0.0834, 0.0562]
IMM = −0.0068 (95% CI: −0.0620, 0.0420)			
Severity infection narrative → Perceived severity infection → Narrative transportation → Protective behavioral intent against COVID-19			
- Infection target: Protagonist’s father	−0.0241	0.0234	[−0.0845, 0.0052]
- Infection target: Narrative’s protagonist	−0.0864	0.0641	[−0.2429, 0.0022]
IMM = −0.0623 (95% CI: −0.1977, 0.0015)			
Severity infection narrative → Perceived severity infection → Cognitive elaboration → Protective behavioral intent against COVID-19			
- Infection target: Protagonist’s father	0.0264	0.0242	[−0.0050, 0.0872]
- **Infection target: Narrative’s protagonist**	**0.0946**	**0.0608**	**[0.0006, 0.2365]**
IMM = 0.0682 (95% CI: 0.0002, 0.1865)			
Severity infection narrative → Perceived severity infection → Reactance → Protective behavioral intent against COVID-19			
- Infection target: Protagonist’s father	0.0192	0.0176	[−0.0023, 0644]
- **Infection target: Narrative’s protagonist**	**0.0690**	**0.0384**	**[0.0079, 0.1557]**
IMM = 0.0497 (95% CI: 0.0048, 0.1189)			

Note: significant specific conditional indirect effects in bold. IMM = index of moderated mediation (difference between conditional indirect effects).

## Data Availability

Datasets and syntax files are available via the Open Science Framework (OSF): https://osf.io/cz3dt/ (accessed on 27 April 2023).

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
