# Peer review of "“It Happened to Me and It’s Serious”: Conditional Indirect Effects of Infection Severity Narrated in Testimonial Tweets on COVID-19 Prevention"

_ijerph, 2023, doi:10.3390/ijerph20136254_

Round 1

Reviewer 1 Report

The work presented in this paper aims to determine which features of testimonial messages are most relevant to increase persuasive impact. An online experiment with a 2 (severity infection narrative: low vs. high) x 2 (infection target: narrative’s protagonist vs. protagonist’s father) between-subjects factorial design was carried out in this work. Young people between 18 to 28 years (N = 278) were randomly assigned to one of the four experimental conditions, where they were asked to read a narrative message in the form of a Twitter thread describing a Covid-19 infection (with mild or severe symptoms) that affected either the protagonist of the message (a 23-year-old young person) or their father. After reading the narrative message the mediating and dependent variables were evaluated. This study highlights that creating persuasive messages based on social media targeted at young people that describe careless behavior resulting in a severe Covid-19 infection can be an appropriate strategy for designing prevention campaigns. The work seems novel. However, parts of the paper need improvement. The following represents my comments/feedback for the improvement of different parts of this paper:

1. The Introduction is more than 5 pages. Please consider creating a separate section on Literature Review to minimize the length of the Introduction section. 

2. Several fact-based statements throughout the paper are missing supporting references. For instance, the statement – “…..the World Health Organisation (WHO) declared a pandemic situation on March 11, 2020” should have a supporting reference.

3. In Figure 2, the severity infection narrative is stated to be in the range of -0.5 to 0.5 However, the graph has the labels -5.0 to 0.5. This seems to be inconsistent.

4. The authors state – “The testimonial message, written in the first person and designed as a Twitter “thread”, featured a young man who indicated having celebrated their birthday by organizing a party with their friends without adopting any preventive measures” The discussion about using Twitter for this work lacks proper justification. Please elaborate on why people used Twitter in the context of COVID-19 with supporting references – for instance: general information seeking and sharing (https://doi.org/10.3390/covid2080076), medical information dissemination (https://doi.org/10.1017/cem.2020.361), etc. to justify using Twitter for the experiment design.

5. Some of the references are missing information. For example, in [43] and [54] it is not clear whether the source is a book/journal/conference paper.

Author Response

Responses to Reviewer 1

Thank you very much for taking the time to review our manuscript and for providing such thoughtful and constructive feedback. We are pleased to hear that you found the paper “novel”. We are also grateful for your kind words and encouragement for our future studies in these important topics. Thank you again for your valuable feedback and for your time in reviewing our manuscript.

Comment # 1 The Introduction is more than 5 pages. Please consider creating a separate section on Literature Review to minimize the length of the Introduction section.

Response:

Following the reviewer’s suggestion, we have split the Introduction from the literature review, now titled, titled “Narrative persuasion with testimonial messages: Advancing in the study of psycho-logical mechanisms”.

Comment # 2 Several fact-based statements throughout the paper are missing supporting references. For instance, the statement – “…..the World Health Organisation (WHO) declared a pandemic situation on March 11, 2020” should have a supporting reference.

Response:

Thank you for bringing this aspect to our attention. We have acknowledged it and included the missing reference.

Comment # 3 In Figure 2, the severity infection narrative is stated to be in the range of -0.5 to 0.5 However, the graph has the labels -5.0 to 0.5. This seems to be inconsistent.

Response:

Thank you for identifying this issue in the Figure “. We have made changes to the Figure 2 to address this problem.

Comment # 4 The authors state – “The testimonial message, written in the first person and designed as a Twitter “thread”, featured a young man who indicated having celebrated their birthday by organizing a party with their friends without adopting any preventive measures” The discussion about using Twitter for this work lacks proper justification. Please elaborate on why people used Twitter in the context of COVID-19 with supporting references – for instance: general information seeking and sharing (https://doi.org/10.3390/covid2080076), medical information dissemination (https://doi.org/10.1017/cem.2020.361), etc. to justify using Twitter for the experiment design.

Response:

Thank you for providing us with two valuable and relevant references to better contextualize our research. However, we have considered that the most suitable place to include these references is at the end of the manuscript, emphasizing the role of Twitter in the dissemination of medical information and misinformation during the COVID-19 pandemic. The manuscript now incorporates those two new references.

Comment # 5 Some of the references are missing information. For example, in [43] and [54] it is not clear whether the source is a book/journal/conference paper.

Response:

Thank you for highlighting this important aspect. We have further enhanced the references by including the ISBN, as they both refer to books.

Reviewer 2 Report

Thank you for the opportunity to review this manuscript.

I truly enjoyed this paper and commend the authors for their research/work.  This is an important topic and I certainly agree with the bias/reactions/etc that go along with social media posts.

The authors do a wonderful job introducing the premise of the research and the 5 hypotheses are very clear.  The results are well discussed.

If I were to suggest anything, it would be to not only mention (which they did) the future potential advantages to support social media readers' behaviors and perceptions surrounding public health initiatives, but also to at least mention that this technique could - and probably is - be used inappropriately as well.

Nice work.

Author Response

Responses to Reviewer 2

Thank you very much for reviewing the manuscript and for the positive feedback on it: “I truly enjoyed this paper and commend the authors for their research/work. This is an important topic and I certainly agree with the bias/reactions/etc that go along with social media posts. The authors do a wonderful job introducing the premise of the research and the 5 hypotheses are very clear. The results are well discussed”.

Comment # 1 If I were to suggest anything, it would be to not only mention (which they did) the future potential advantages to support social media readers' behaviors and perceptions surrounding public health initiatives, but also to at least mention that this technique could - and probably is - be used inappropriately as well.

Response:

Thank you for this insightful contribution, which we fully agree with. Furthermore, it echoes some of the points raised by Reviewer 1. For this reason, and based on two new references we have incorporated into the manuscript, we have added a sentence in the conclusions to briefly mention the opportunities as well as the risks of disseminating health information through social media platforms, specifically Twitter. Considering this, and with reference to two newly added references in the manuscript, we have included a sentence in the Conclusions to briefly address the opportunities and risks associated with disseminating healthcare information through social media, particularly Twitter.

Reviewer 3 Report

This paper explores a very interesting and pertinent topic in the field of health communication. 

In structural terms, the article is well structured, with a clear and consistent methodological design, making relevant contributions to scientific knowledge in this area, demonstrating the importance of communication in health and in the prevention of risk behaviours.

The only aspect that I would advise to do is to reduce the number of key words to five, at most six.

Author Response

Responses to Reviewer 3

Firstly, we greatly appreciate the positive feedback provided on our manuscript: “This paper explores a very interesting and pertinent topic in the field of health communication. In structural terms, the article is well structured, with a clear and consistent methodological design, making relevant contributions to scientific knowledge in this area, demonstrating the importance of communication in health and in the prevention of risk behaviours”.

Comment # 1 The only aspect that I would advise to do is to reduce the number of key words to five, at most six.

Response:

In light of the reviewer's feedback, we have addressed their comment by removing two keywords, resulting in a revised set of six keywords now being included.

Round 2

Reviewer 1 Report

The authors have revised their paper as per all my comments and feedback. I do not have any additional comments at this point. I recommend the publication of the paper in its current form.